# Comprehensive Use of Routine Clinical Parameters to Identify Patients at Risk of New-Onset Atrial Fibrillation in Acute Myocardial Infarction

**DOI:** 10.3390/jcm10163622

**Published:** 2021-08-17

**Authors:** Monika Raczkowska-Golanko, Grzegorz Raczak, Marcin Gruchała, Ludmiła Daniłowicz-Szymanowicz

**Affiliations:** 1Department of Cardiology and Electrotherapy, Medical University of Gdańsk, 80-210 Gdańsk, Poland; mrg@gumed.edu.pl (M.R.-G.); grzegorz.raczak@gumed.edu.pl (G.R.); 2I Department of Cardiology, Medical University of Gdańsk, 80-210 Gdańsk, Poland; marcin.gruchala@gumed.edu.pl

**Keywords:** new-onset atrial fibrillation (NOAF), atrial fibrillation, acute myocardial infarction

## Abstract

(1) Background: New-onset atrial fibrillation (NOAF) is a significant complication of acute myocardial infarction (AMI). Our study aimed to investigate whether routinely checked clinical parameters aid in NOAF identification in modernly treated AMI patients. (2) Patients and methods: Patients admitted consecutively within 2017 and 2018 to the University Clinical Centre in Gdańsk (Poland) with AMI diagnosis (necrosis evidence in a clinical setting consistent with acute myocardial ischemia) were enrolled. Medical history and clinical parameters were checked during NOAF prediction. (3) Results: NOAF was diagnosed in 106 (11%) of 954 patients and was significantly associated with in-hospital mortality (OR 4.54, 95% CI 2.50–8.33, *p* < 0.001). Age, B-type natriuretic peptide (BNP), C-reactive protein (CRP), high-sensitivity troponin I, total cholesterol, low-density lipoprotein cholesterol, potassium, hemoglobin, leucocytes, neutrophil/lymphocyte ratio, left atrium size, and left ventricular ejection fraction (LVEF) were associated with NOAF in the univariate logistic analysis, whereas age ≥ 66 yo, BNP ≥ 340 pg/mL, CRP ≥ 7.7 mg/L, and LVEF ≤ 44% were associated with NOAF in the multivariate analysis. (4) Conclusions: NOAF is a multifactorial, significant complication of AMI, leading to a worse prognosis. Simple, routinely checked clinical parameters could be helpful indices of this arrhythmia in current invasively treated patients with AMI.

## 1. Introduction

Atrial fibrillation (AF) is the most common arrhythmia [1], present in approximately 0.4% of the general population and 2–4% of patients over 60 years old [2]. It often complicates acute myocardial infarction (AMI), with an incidence documented between 5 and 22% [3,4]. This arrhythmia is closely associated with prolonged hospitalization, higher in-hospital mortality, and worse outcomes [5,6]. The clinical significance and management of new-onset atrial fibrillation (NOAF) in patients with AMI is frequently debated and not fully understood; therefore, identifying risk factors related to NOAF in AMI is still of great clinical value. Most previous studies usually prioritized only one clinical parameter [7,8,9,10,11,12,13,14,15] or considered only patients with one type of AMI, mostly ST-Elevation Myocardial Infarction (STEMI) [9,16,17,18,19,20]. Some studies were based on international registers [21,22,23,24], which, beyond the obvious advantages, included data from different clinical centers, sometimes from different countries, which could implicate different diagnostic methods and different treatment possibilities [11,25]; most of them were performed in the earlier years, based upon the previous guidelines of AMI treatment. Our study aimed to check whether the comprehensive use of the routinely checked clinical parameters could help to identify patients with a high probability of NOAF based on current, consecutive AMI patients hospitalized and treated according to the current guidelines in one large clinical center.

## 2. Materials and Methods

Our study retrospectively included all patients hospitalized with an AMI diagnosis in the University Clinical Centre in Gdańsk (Poland) from January 2017 to December 2018. The data were collected through MedStream Designer, which was fully integrated with the hospital information system. The exclusion criterion was age younger than 18 years. AMI was diagnosed based on the appropriate measures [26,27]. AMI was diagnosed if there was evidence of myocardial injury (defined as elevation of cardiac troponin values with at least one value above the 99th percentile upper reference limit) with necrosis in a clinical setting consistent with myocardial ischemia. Diagnosis of AF was based on the physician’s interpretation of ECG according to the current guidelines [28]; all patients were 24 h monitored (with the possibility of the retrospective analysis of the ECG) during their hospital stay in the intensive cardiac care unit or monitoring room in the regular cardiac ward; afterward, they had 12-lead ECG performed daily during their hospital admission or when any new symptoms were noticed; any observations of rhythm changes were registered. The term NOAF was considered for any newly diagnosed AF (absence of P waves, atrial activity represented by fibrillatory waves, and irregular RR intervals) that appeared during the index hospitalization, which lasted at least 30 s or entire 12-lead ECG. Medical history (with particular attention to coronary artery disease, including myocardial infarction (MI) and revascularizations, and others), echocardiography, laboratory parameters at admission, and pharmacological and invasive treatment within the hospitalization were taken into consideration and compared between the patients with (NOAF group) and without (non-NOAF) this arrhythmia.

Coronary angiography and percutaneous coronary intervention (PCI) within hospitalization were performed according to the newest guidelines [26,27]; the results of angiograms were graded as to the number of diseased coronary arteries; a coronary artery was considered diseased if there was any obstructive lesion ≥ 50% in diameter in the left main stem, ≥70% in a major coronary vessel, or 30% to 70% stenosis with fractional flow reserve ≤ 0.8. Coronary blood flow assessed during PCI was determined according to Thrombolysis in Myocardial Infarction (TIMI) classification. Additionally, data regarding the length of hospitalization, in-hospital mortality, and post-discharge medical treatment were analyzed. The selection of antithrombotic therapy at discharge was at the discretion of the attending physician.

The Independent Bioethical Committee approved the study’s protocol for Scientific Research of the Medical University of Gdansk (consent number NBBN/290/2018). This was a retrospective study of data routinely collected in clinical practice; therefore, the requirement for written and informed consent was waived.

### Statistical Analysis

Continuous data are presented as median (25th–75th percentile) and categorical data as numbers (n) and percentages (%). We performed the Shapiro–Wilk test to determine whether our data were normally distributed; most of the analyzed parameters did not have a normal data distribution, even after logarithmic transformation; therefore, we selected appropriate statistical analysis methods based on non-parametric tests. Comparisons between groups were performed with the Mann–Whitney U-test for continuous variables and Pearson’s chi-square test for categorical variables, as appropriate. The predictability of the established variables as potential predictors of NOAF was determined by the area (AUC) under the receiver operating characteristic (ROC) curve; adequate cut-off values were identified according to the best paring of sensitivity and specificity values. Logistic regression analyses were performed to detect which parameters (with pre-specified cut-off values) showed the most substantial relation to the NOAF (univariate analyses). Multivariate analysis was applied to continuous data (dichotomized according to the cut-off values identified in ROC analyses) and categorical data significantly associated in the univariate analyses with NOAF (*p*-value of 0.05 or less); the set of variables accepted for the model was determined by the backward elimination method from the setting of all statistically significant predictors. Values of *p* < 0.05 were considered significant. The statistical analysis was conducted using the R 3.1.2 environment (R Core Team, Vienna, Austria). 

## 3. Results

### 3.1. Baseline Clinical Characteristics 

A total of 954 AMI patients were enrolled in the study. Of these patients, 106 (11%) were diagnosed with NOAF. Amongst 106 NOAF patients, the majority (66 patients—62%) had arrhythmia diagnosed within the first day of hospitalization, and 19 (18%) had NOAF at the time of admission. In addition, patients who developed NOAF were older and had a lower body mass index (BMI), without other baseline clinical differences (Table 1).

### 3.2. Laboratory and Echocardiographic Parameters

There were many significant differences in laboratory and echocardiography results between NOAF and non-NOAF patients (Table 2). At admission, patients with NOAF had a higher level of brain natriuretic peptide (BNP), C-reactive protein (CRP), leucocyte, and high-sensitivity troponin I (hsTnI). Although the free thyroxine (FT4) was within the normal range, the level was significantly higher in the NOAF patients. In addition, sodium, potassium, total cholesterol (TC), low-density lipoprotein cholesterol (LDL-C), and hemoglobin were significantly lower. Regarding echocardiography parameters, patients with NOAF had significantly lower left ventricular ejection fraction (LVEF) and left atrium (LA) size, as well as worse right ventricular (RV) parameters such as RV internal dimension (RVID) and tricuspid annular plane systolic excursion (TAPSE). 

### 3.3. Percutaneous Coronary Interventions

The compared group did not differ in STEMI and non-STEMI types of AMI (Table 3). The majority of patients (97%) had coronary angiography during hospitalization, whereas 82% had a percutaneous coronary intervention. In addition, most of the patients had successful intervention (98% had TIMI flow 3), with no differences in angiographic results between the NOAF and non-NOAF groups. 

### 3.4. Predictors of NOAF 

ROC analysis identified BNP with the cut-off value 340 pg/mL as the most accurate predictor of NOAF (AUC 70.5% [64.6–76.5%]). The rest of the parameters were characterized by lower discriminatory power (Table 4 presents parameters with AUC higher than 50% in ROC analysis). 

Univariate logistic regression analyses revealed age, length of hospitalization, BNP, hsTnI, CRP, potassium, hemoglobin, leucocytes, neutrophil to lymphocyte ratio, LDL-C, total cholesterol, creatinine, LA size, and LVEF with pre-specified cut-off values as significant predictors of NOAF (Figure 1).

Age ≥ 66 yo, BNP ≥ 340 pg/mL, CRP ≥ 7.7 mg/L, and LVEF ≤ 44% maintained their significance in NOAF prediction in multivariate analysis; BNP was found to be the parameter with the highest predictive power (Table 5).

### 3.5. Outcomes

Patients with NOAF had more prolonged hospitalizations, more in-hospital adverse events, and worse in-hospital prognosis (Table 6). In addition, NOAF was found to be significantly associated with in-hospital mortality (OR 4.54 [95% CI 2.50–8.33], *p* < 0.001).

Only 74 of 87 surviving NOAF patients (85%) were discharged with sinus rhythm, which was a significantly lower rate than in the non-NOAF group (92%—745 of 809), *p* < 0.001.

### 3.6. Pharmacological Treatment at Discharge

Due to the retrospective nature of the study, information about pharmacological treatment for four patients was absent; therefore, further analyses were performed for 892 patients. The main difference between the groups that received antithrombotic therapy was a significantly higher rate of NOACs, but lower aspirin and ticagrelor were noticed in the NOAF group. The frequencies of other medications (beta-blockers, angiotensin-converting enzyme inhibitors, angiotensin II receptor blockers, statins) were not statistically different between the groups (Table 7).

## 4. Discussion

Our study revealed that that in the modern revascularization era (when most of the patients with AMI are successfully treated invasively), routinely checked clinical parameters could help to identify those at risk of NOAF among consecutive patients, regardless of the type of infarction. To the best of our knowledge, this is the first study based on current patients with AMI, where a complex evaluation of the routinely checked clinical parameters based on the data from a high-volume tertiary care center was performed.

### 4.1. Clinical, Laboratory, and Echocardiographic Parameters

Data from the literature postulate that 5 to 22% of AMI patients have NOAF during their acute hospitalization [6,29,30]. In agreement with data from the literature, our patients with NOAF were older [6,13,23,31], and age was an independent predictor of this arrhythmia [22,32,33,34]. Among laboratory parameters, we revealed some statistical differences in NOAF compared to the non-NOAF group. Some of them are well-known predictors of AF, but there are some discrepancies in the literature regarding others. Moreover, some parameters are presented as NOAF risk factors for the first time in the literature.

One of the well-known parameters connected to AF development in the general population is low potassium level in serum [35,36,37,38]. In the Rotterdam Study [35], potassium below 3.50 mmol/L was associated with a higher risk of this arrhythmia. Campbell et al. [37] investigated the impact of maintaining serum potassium ≥ 3.6 mmol/L in comparison to ≥4.5 mmol/L on the incidence of NOAF after coronary artery bypass grafting, but the authors did not publish the results. In another study [36], preoperative hypokalemia (<3.5 mmol/L) was associated with AF. None of the abovementioned data concerned patients with AMI. In our study, the median potassium value was within the normal range in both groups; however, for NOAF patients, it was significantly lower. In our study, the cut-off value for potassium calculated in the ROC analysis of 4.2 mmol/L was found to be crucial in revealing the NOAF probability, which is one of the novelties of our research.

Another predictor of NOAF in our results was hemoglobin. There is some discrepancy in the literature concerning this parameter. For instance, Distelmaier et al. [10] demonstrated that patients with AF in the setting of AMI displayed significantly higher levels of hemoglobin. However, other data [39,40] suggest that NOAF onset is associated with a lower hemoglobin level. Our results are in line with the latter findings. A potential explanation of the connection between AF and a lower hemoglobin level might be that anemia causes decreased oxygen-carrying capacity, increasing cardiac output to maintain tissue oxygen delivery [41]. Moreover, increased neurohormonal activity in anemia can cause arrhythmogenic remodeling susceptible to AF [40]. Our results postulate that not anemia but a hemoglobin level below 14 mg/dL in AMI patients could indicate a risk of NOAF development. This is another novelty of our study.

Troponin concentration as a marker of AMI intensity is another parameter that could influence AF. Data from the Framingham Heart Study [42] and the Atherosclerosis Risk in Communities (ARIC) study [43] revealed a prognostic role of troponins in AF prediction in a ten-year follow-up observation in the general population. Moreover, Parashar et al. [21] disproved the relationship between the level of troponins and the occurrence of NOAF, but his study did not concern AMI patients. Later data from the Busselton health study proved that elevated troponin levels could be an independent predictor of hospitalization due to AF [44]. For the first time in the literature, our study confirms the hsTnI level’s significance in NOAF prediction in AMI, but only in the univariate logistic regression analysis.

Dyslipidemia is a significant factor in the development of coronary heart disease and atherosclerosis [45], whereas its role in the development of AF is less clear. Annoura et al. reported the “cholesterol paradox” in AF patients and found lower serum cholesterol levels and triglycerides in patients with paroxysmal AF [46]. Another study [25] showed that low serum levels of LDL-C and high-density lipoprotein cholesterol (HDL-C) were present in patients with AF, irrespective of the type of AF. Watanabe et al. [47] demonstrated that a low HDL-C level was strongly associated with an increased risk of developing AF, and the total cholesterol and LDL-C levels were contrarily associated with AF. In our study, total cholesterol and LDL-C were significantly lower in the NOAF group, which is in line with the previous literature.

Inflammation is a known process that can lead to atrial structural and electrical remodeling, predisposing patients to AF [12]. Extensive inflammation in patients with AMI can lead to the development of NOAF [11,46,48,49,50]. Moreover, inflammation has a fundamental role in atherosclerotic plaque rupture and seems to play an important role in the prothrombotic state associated with AF [51,52]. In our study, leucocytes, the neutrophil to lymphocyte ratio, and CRP as inflammation parameters were found to be significantly associated with NOAF in the univariate logistic regression analysis. Furthermore, CRP ≥ 7.7 mg/L, contrary to other inflammatory parameters and other abovementioned parameters connected to NOAF, displayed significance in the multivariate analysis.

A vital laboratory parameter that was found to be significant for revealing NOAF not only in the univariate but in the multivariate analysis in our study was BNP. BNP is a hormone secreted predominantly by the ventricles and increases markedly in patients with congestive heart failure in proportion to its severity [53]. According to data from the literature, an increased BNP level is reported in the first 24 h after AMI, revealing the compensatory role of ventricular dysfunction caused by AMI, reducing progressive ventricular enlargement and attenuating ventricular remodeling after AMI [54]. Moreover, data from the TRIUMPH registry showed that elevated BNP predicts NOAF in AMI patients [21]. Asanin et al. [9] demonstrated that BNP might be involved in the risk prediction of NOAF in the setting of STEMI treated by primary PCI, with BNP level ≥ 720 pg/mL as the most potent predictive factor. A lower cut-off value of BNP (263 pg/mL) was demonstrated in another study also for NOAF after STEMI [13]. We confirmed this observation in our research based on all AMI patients, not only those with STEMI. We proved that BNP with a cut-off value of ≥340 pg/mL is a significant, independent predictor of NOAF in the setting of AMI, which is consistent with previous findings [9,13].

It is well known that the probability of AF increases with the enlargement of LA and reduction in LVEF [6,55,56,57]. The Cardiovascular Health Study [58] proved more than a double-fold increase in the development of NOAF when the LA diameter is more than 40 mm. The GUSTO-I trial [30] demonstrated LVEF with a cut-off value of 42.7% as a predictor of NOAF in AMI patients. According to a meta-analysis conducted in 2017, including ten studies comprising a total of 708 NOAF patients and 6785 controls, both decreased LVEF and increased LA levels were associated with greater risk of NOAF following AMI [59]. Notably, the three most extensive studies revealed LVEF < 45% as an independent predictor of AF [8,49,60]. Our study is in line with these results: LA ≥ 41 mm and LVEF ≤ 44% were significant predictors of NOAF in the univariate logistic analysis in AMI patients; furthermore, LVEF remained significant in the multivariate analysis.

### 4.2. Prognosis of NOAF Patients

Many authors have intensively studied the influence of NOAF on prognosis [21,22,61,62]. An extensive meta-analysis from 2011, based on 43 studies involving 278,854 patients, showed that AF in AMI is associated with at least a 40% increase in mortality compared to patients with sinus rhythm [61]. Data from the literature have also demonstrated malignant ventricular arrhythmias (ventricular tachycardia and fibrillation) and complete atrioventricular blocks as more frequent complications in patients with NOAF [22,30,63]. Our data are in line with the previous results: we revealed that NOAF increased the risk of in-hospital mortality in AMI patients more than four-fold (OR 4.54 [95% CI 2.50–8.33], *p* < 0.001), and the frequency of life-threatening arrhythmias was higher in these patients.

### 4.3. Pharmacological Treatment

Antithrombotic therapy is the most important therapy in reducing the burden of stroke and death in patients with AF, including NOAF [61,64,65,66]. Our study shows a high rate of recommended anticoagulation at discharge: more than 70% of patients diagnosed with NOAF received oral anticoagulation at discharge, and most of them were on NOACs (63%). The worse compliance with recommendations in our population concerns triple antithrombotic treatment (oral anticoagulation and dual antiplatelet therapy), which, according to the most recent guidelines, should be used in every patient with AF undergoing a primary PCI for AMI [26,27]. As we show in Table 3, only 57% of NOAF patients were on triple antithrombotic therapy at discharge. Fortunately, this is a higher rate than described in the literature some years ago [22,64,67]. The disproportion between the guidelines and the real-life rate of triple antithrombotic treatment in NOAF patients could have several clinical explanations. On the one hand, the high risk of bleeding or bleeding-related complications could influence the physicians’ decision to leave the patient on the double or even single antithrombotic therapy. On the other hand, there are no guidelines concerning precise information about the treatment of NOAF patients. Axelrod et al. [29] suggested that “early-paroxysmal AF” that resolved within 24 h of admission may not have a high stroke risk, questioning the indication for long-term anticoagulation contrary to “late-AF” beyond the first 24 h, which should be treated appropriately. These data need to be established in further research due to their potentially high clinical importance.

### 4.4. Novelties of the Study

The present study differs from similar previous ones in several features. Our study was based on a large group of current (treated in 2017 and 2018) European AMI patients, including not only STEMI but NSTEMI patients as well. Our study was based on all consecutively admitted patients with a high rate of invasive treatment regardless of the type of infarction: 97% of patients had coronary angiography, and 82% of patients had PCI, in contrast to older studies [21,68]. Huge European registries were performed in earlier years based upon older guidelines [22,23,24]; some current registries include patients of various ethnicities [21].

In our recently treated group of patients, we focused on studying not only one, as in many other similar studies, but on many routinely checked clinical parameters in the prediction of NOAF. It is worth stressing that our analysis was based on all consecutive AMI patients, regardless of the type of infarction. We calculated cut-off values for these parameters. Age, length of hospitalization, BNP, hsTnI, CRP, potassium, hemoglobin, leucocytes, neutrophil to leucocyte ratio, TC, LDL-C, creatinine, LA size, and LVEF were found to be significant in the univariate logistic regression analysis, whereas age, BNP, CRP, and LVEF were independent indices of this arrhythmia in the multivariate analysis. The possibility of the prediction NOAF in current, recently treated AMI patients may have important clinical implications, allowing for more careful monitoring of such patients, longer ECG observation (more than 24 h), some laboratory parameters (mainly BNP and CRP), and precise measurement of LVEF within hospitalization and after discharge. This was not the subject of this study but requires further research, mainly since the guidelines covering ACS or AF describe a group of patients with NOAF to a minimal extent [28,69].

As we mentioned above, our group had a higher rate of triple antithrombotic treatment regarding NOAF patients than in previous studies [22,64,67]. However, this is still insufficient in order to maintain the guidelines. The main question of how to determine the optimal anticoagulation therapy in NOAF patients with AMI remains for future investigation. Our findings can improve the determination of patients that may develop NOAF in the setting of AMI and therefore need to be treated appropriately.

## 5. Limitations

Our study has several limitations. Firstly, this is a retrospective investigation, limited to available data and parameters in patients’ medical records. Only patients with AMI (patients with evidence of myocardial injury with necrosis in a clinical setting consistent with myocardial ischemia) were included, and we could not include patients with unstable angina; therefore, our results could not be applied to all patients with acute coronary syndrome. Secondly, some patients with AF on admission may have had previous yet undiagnosed paroxysmal AF and new-onset arrhythmias, leading to an overestimated incidence. However, the 24 h monitoring in the intensive cardiac care unit and querying of symptoms during medical visits were achieved routinely during hospitalization; the actual incidence of NOAF may have been underestimated because of asymptomatic AF episodes. Since we do not have detailed data on the duration of AF, we cannot comment on cause and effect. There is a possibility that NOAF may lead to higher levels of biomarkers or higher levels of biomarkers in the setting of AMI may lead to NOAF. Moreover, we could not precisely define the time of VT/VF/AVB during hospitalization; therefore, some patients with these complications could have had malignant arrhythmias at the time of admission, not only during their hospitalization. Finally, our study was a single-center study, which is another limitation; however, some benefits associated with the single-center nature of the study could be identified (including laboratory and echocardiography data collected from the same laboratory, obtained mainly by the same experts, decreasing interobserver variability).

## 6. Conclusions

This study shows that in the era of modern revascularization, new-onset atrial fibrillation remains a frequent complication of acute myocardial infarction and is associated with higher in-hospital mortality. Older age and routinely checked parameters, such as higher BNP and CRP levels, and lower LVEF could be helpful indices of this arrhythmia in current, mostly invasively treated patients with AMI.

## Figures and Tables

**Figure 1 jcm-10-03622-f001:**
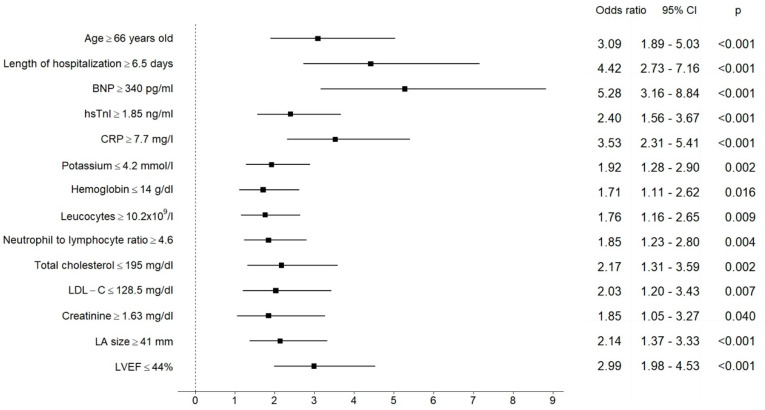
Results of univariate logistic regression analysis for the pre-specified cut-off values of analyzed parameters as predictors of NOAF. The central estimate and 95% confidence interval for odds ratio are shown.

**Table 1 jcm-10-03622-t001:** Baseline clinical characteristics.

	All Patients *n* = 954	NOAF *n* = 106	Non-NOAF *n* = 848	*p*
Age (years old)	69 (61–78)	74 (66–84)	67 (60–76)	<0.001
Male sex, *n* (%)	637 (67%)	67 (63%)	571 (67%)	0.444
BMI (kg/m^2^)	28 (25–31)	27 (24–30)	28 (25–31)	0.027
Prior MI, *n* (%)	276 (29%)	31 (29%)	245 (29%)	0.999
Prior revascularization (PCI/CABG), *n* (%)	270 (28%)	26 (25%)	244 (29%)	0.424
Hypertension, *n* (%)	719 (75%)	79 (75%)	640 (76%)	0.812
Diabetes mellitus, *n* (%)	314 (33%)	31 (29%)	283 (33%)	0.812
Previous stroke, *n* (%)	70 (7%)	10 (9%)	60 (7%)	0.427
Pacemaker, *n* (%)	29 (3%)	5 (5%)	24 (3%)	0.360
ICD, *n* (%)	35 (4%)	3 (3%)	32 (4%)	0.789
On-admission treatment				
Aspirin, *n* (%)	357 (38%)	43 (41%)	314 (37%)	0.524
Angiotensin converting enzyme inhibitors, angiotensin receptor blockers, *n* (%)	511 (54%)	53 (50%)	458 (54%)	0.470
Statins, *n* (%)	376 (40%)	41 (39%)	335 (40%)	0.916

BMI—body max index; CABG—coronary artery bypass grafting; ICD—implantable cardioverter-defibrillator; MI—myocardial infarction; PCI—percutaneous coronary intervention.

**Table 2 jcm-10-03622-t002:** Laboratory and echocardiographic parameters.

	All Patients (*n* = 954)	NOAF (*n* = 106)	Non-NOAF (*n* = 848)	*p*
Laboratory parameters				
BNP, pg/mL	512 (59–541)	791 (193–1087)	471 (54–429)	<0.001
hsTnI, ng/mL	4.89 (0.05–1.42)	6.41 (0.06–4.84)	4.70 (0.04–1.20)	0.020
CK-MB, ng/mL	19.2 (2.1–10.3)	18.42 (2.2–14)	19.30 (2.1–10)	0.167
CRP, mg/L	18.9 (1.6–14.2)	36.1 (3.3–36.2)	16.6 (1.5–11.3)	<0.001
Sodium, mmol/L	138 (136–140)	137 (135–140)	138 (136–140)	0.033
Potassium, mmol/L	4.3 (4–4.6)	4.2 (3.8–4.6)	4.35 (4–4.6)	0.008
Hemoglobin, g/dL	13.5 (12.4–15)	13.3 (12–14.9)	13.6 (12.4–15)	0.042
Leucocytes, 10^9^/L	10.61 (7.82–12.48)	11.91 (8.07–13.91)	10.45 (7.80–12.34)	0.015
Neutrophil to lymphocyte ratio	5.3 (2.1–5.6)	6.0 (2.2–6.9)	5.3 (2.1–5.4)	0.051
Total cholesterol, mg/dL	180 (143–214)	165 (129–192)	182 (144–217)	0.005
LDL-C, mg/dL	110 (77–141)	98 (64–125)	112 (79–144)	0.011
Creatinine, ml/dL	1.17 (0.78–1.19)	1.14 (0.77–1.25)	1.17 (0.78–1.18)	0.225
TSH, uU/L	1.494 (0.577–1.721)	1.29 (0.66–1.86)	1.53 (0.52–1.70)	0.233
FT3, pmol/L	3.52 (2.63–3.70)	3.14 (2.74–3.41)	3.58 (2.60–3.70)	0.334
FT4, pmol/L	13.50 (11.68–14.84)	14.67 (13.26–16.20)	13.34 (11.55–14.74)	0.005
Glucose, mg/dL	159 (104–178)	184 (120–219)	156 (103–173)	<0.001
Echocardiographic parameters				
LA size, mm	40 (36–44)	43 (38–46)	40 (36–44)	<0.001
LVIDd, mm	50 (45–54)	51 (44–55)	50 (45–54)	0.208
LVIDs, mm	35 (30–39)	38 (31–44)	35 (29–39)	<0.001
LVEF, %	47 (40–56)	42 (32–51)	48 (40–57)	<0.001
RVID, mm	37 (32–41)	40 (34–44)	37 (32–40)	0.006
TAPSE, mm	20 (17–23)	18 (14–22)	20 (17–24)	0.003
RVSP, mmHg	41 (34–47)	42 (35–48)	40 (32–47)	0.277
Mitral regurgitation				
Moderate, *n* (%)	176 (25%)	29 (28%)	147 (24%)	0.082
Severe, *n* (%)	43 (6%)	7 (7%)	36 (6%)	

BNP—B-type natriuretic peptide; CK-MB—creatine kinase muscle-brain; CRP—C-reactive protein; FT3—free triiodothyronine; FT4—free thyroxine; hsTnI—high sensitivity troponin I; LA—left atrium; LDL-C—low-density lipoprotein cholesterol; left ventricular ejection fraction—LVEF; LVIDd—left ventricular internal diameter end diastole; LVIDs—left ventricular internal diameter end systole; LVEF—left ventricular ejection fraction; RVIDd—right ventricular internal dimension; RVSP—right ventricular systolic pressure; TAPSE—tricuspid annular plane systolic excursion; TSH—thyroid-stimulating hormone.

**Table 3 jcm-10-03622-t003:** Types of myocardial infarction, results of coronary angiography, and effects of PCI.

	All Patients (*n* = 954)	NOAF (*n* = 106)	Non-NOAF (*n* = 848)	*p*
Types of myocardial infarction				
ST-elevation MI, *n* (%)	327 (34%)	42 (40%)	285 (34%)	0.233
Non-ST-elevation MI, *n* (%)	627 (66%)	64 (60%)	563 (66%)	0.233
Results of coronary angiography with the number of stenotic vessels				
In-hospital coronary angiography, *n* (%)	921 (97%)	99 (93%)	822 (97%)	0.083
Patients with PCI	779 (82%)	81 (76%)	698 (82%)	0.522
Results of coronary angiography—significant stenosis				
One vessel, *n* (%)	313 (34%)	33 (33%)	280 (35%)	0.317
Two vessels, *n* (%)	264 (29%)	25 (25%)	239 (29%)	
Multivessel disease, *n* (%)	286 (31%)	33 (33%)	253 (31%)	
None, *n* (%)	49 (5%)	9 (9%)	40 (5%)	
PCI effects				
TIMI flow 1	2 (0.3%)	0 (0%)	2 (0.2%)	0.522
TIMI flow 2	13 (1.7%)	0 (0%)	13 (1.8%)	
TIMI flow 3	764 (98%)	81 (100%)	683 (98%)	

MI—myocardial infarction; PCI—percutaneous coronary intervention; TIMI—Thrombolysis In Myocardial Infarction.

**Table 4 jcm-10-03622-t004:** ROC analysis with cut-off values of the analyzed parameters as NOAF predictors.

	Cut-Off Values	AUC
Age	≥66 years old	65.4% (60.1–70.8%)
Length of hospitalization	≥6.5 days	67.8% (62.0–73.6%)
BNP	≥340 pg/mL	70.5% (64.6–76.5%)
hsTnI	≥1.85 ng/mL	57.0% (50.9–63.1%)
CRP	≥7.7 mg/L	66.1% (60.2–71.9%)
Potassium	≤4.2 mmol/L	57.9% (51.5–64.3%)
Hemoglobin	≤14 g/dL	55.2% (49.2–61.1%)
Leucocytes	≥10.2 × 10^9^/L	57.3% (51.1–63.4%)
Neutrophil to lymphocyte ratio	≥4.6	57.9% (48.4–67.5%)
Total cholesterol	≤195 mg/dL	58.6% (52.7–64.6%)
LDL-C	≤128.5 mg/dL	56.8% (50.7–62.9%)
Creatinine	≥1.63 mL/dL	52.3% (46.3–58.2%)
LA size	≥41 mm	62.0% (56.0–68.0%)
LVEF	≤44 %	64.3% (58.6–70.1%)

AUC—area under the receiver-operating characteristic (ROC) curve; BNP—B-type natriuretic peptide; CRP—C-reactive protein; hsTnI—high sensitivity troponin I; LA—left atrium; LDL-C—low-density lipoprotein cholesterol; LVEF—left ventricular ejection fraction.

**Table 5 jcm-10-03622-t005:** Significant predictors of NOAF in multivariate logistic regression analysis.

Cut-Off Value	OR (95% CI)	*p*
Age ≥ 66 years old	2.37 (1.23–4.58)	0.009
BNP ≥ 340 pg/mL	4.60 (2.27–9.32)	0.004
CRP ≥ 7.7 mg/L	2.02 (1.14–3.56)	0.010
LVEF ≤ 44%	1.93 (1.12–3.12)	0.020

BNP—B-type natriuretic peptide; CRP—C-reactive protein; LVEF—left ventricular ejection fraction. The multivariate model was determined by the backward elimination method from the setting of all parameters significantly predicted NOAF in univariate analysis (presented in Figure 1).

**Table 6 jcm-10-03622-t006:** In-hospital prognosis.

	All Patients (*n* = 954)	NOAF (*n* = 106)	Non-NOAF (*n* = 848)	*p*
Length of hospitalization (days)	10 (5–11)	14 (7–17)	9 (5–9)	<0.001
VF during hospitalization, *n* (%)	65 (7%)	14 (13%)	51 (6%)	0.012
VT during hospitalization, *n* (%)	26 (3%)	6 (6%)	20 (2%)	0.059
AVB III during hospitalization, *n* (%)	15 (2%)	6 (6%)	9 (1%)	0.004
Stroke during hospitalization, *n* (%)	9 (1%)	3 (3%)	6 (1%)	0.068
In-hospital mortality, *n* (%)	58 (6%)	19 (18%)	39 (5%)	<0.001

AVB—atrioventricular block; SR—sinus rhythm; VF—ventricular fibrillation; VT—ventricular tachycardia.

**Table 7 jcm-10-03622-t007:** Pharmacological treatment at discharge.

	All Patients (*n* = 892)	NOAF (*n* = 86)	Non-NOAF (*n* = 806)	*p*
Beta-blockers, *n* (%)	776 (87%)	76 (88%)	700 (87%)	0.866
ACE inhibitors/ARBs, *n* (%)	802 (90%)	73 (85%)	729 (91%)	0.127
Statins, *n* (%)	842 (94%)	81 (94%)	761 (94%)	0.809
Antithrombotic therapy				
Aspirin, *n* (%)	843 (94%)	76 (88%)	767 (95%)	0.020
Clopidogrel, *n* (%)	691 (77%)	72 (84%)	619 (77%)	0.174
Ticagrelor, *n* (%)	148 (17%)	3 (3%)	145 (18%)	<0.001
Vitamin K antagonists, *n* (%)	55 (6%)	8 (9%)	47 (6%)	0.233
NOACs, *n* (%)	141 (16%)	54 (63%)	87 (11%)	<0.001
Low-molecular-weight heparins, *n* (%)	42 (5%)	7 (8%)	35 (4%)	0.173
Triple antithrombotic therapy				
Aspirin + Clopidogrel + Vitamin K antagonists	46 (5.1%)	8 (9%)	38 (4.7%)	<0.001
Aspirin + Clopidogrel + NOACs	108 (12.1%)	40 (47%)	68 (8.4%)	
Aspirin + Clopidogrel + LMWH	3 (0.3%)	1 (1%)	2 (0.2%)	
Double antithrombotic therapy				
Aspirin + Clopidogrel	491 (55%)	14 (16%)	477 (59%)	<0.001
Aspirin + Ticagrelor	139 (16%)	2 (2%)	137 (17%)	

ACE—angiotensin-converting enzyme; ARBs—angiotensin receptor blockers; LMWH—low-molecular-weight heparin; NOACs—novel oral anticoagulants.

## Data Availability

Data are available on request due to privacy and ethical restrictions.

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
