# Peer review of "Comprehensive Use of Routine Clinical Parameters to Identify Patients at Risk of New-Onset Atrial Fibrillation in Acute Myocardial Infarction"

_jcm, 2021, doi:10.3390/jcm10163622_

Round 1

Reviewer 1 Report

Raczkovska-Golanko and colleagues presented the original work, emphasizing how routine measurement of clinical parameters could be useful in assessing new-onset atrial fibrillation (NOAF) in patients with acute myocardial infarction (AMI). This topic is important given the high association of NOAF with AMI complications and in-hospital mortality. The manuscript is well written and structured, the results are clearly presented. Authors well-discussed literature data, citing both papers that confirm and refute their assumptions, trying to present their point of view and explanations. This work is certainly well developed and could be of interest to readers. 

Author Response

Dear Reviewer,

We would like to express our thanks for a thorough and detailed revision of our manuscript. We hope that you will find our paper interesting and you will approve it for publication. 

We tried to check the English language and style and correct all the mistakes and hypos using a qualified English-speaking person. We are ready for further actions in this area whenever necessary.

All changes to the manuscript text are tracked in the revised manuscript file with the Track Changes' mode.

Yours faithfully,

Prof. Ludmiła Daniłowicz-Szymanowicz, MD PhD
Monika Raczkowska-Golanko, MD

Reviewer 2 Report

Dear authors,

I have read your manuscript with great interest, as predicting AF in AMI is vital to stratify this very complex patient collective in terms of their overall risk and outcome. However, a few points still need explaining:

-) Title: Maybe shorten it to: “Comprehensive use of routine clinical parameters to identify patients at risk of new-onset atrial fibrillation in acute coronary syndrome” (or myocardial infarction, see comment #2)

-) Please use the most up-to-date wording for acute coronary syndrome following the latest ESC guidelines.

-) Abstract, 2nd sentence: This statement is very vague. I would either give more precise information or delete it.

-) Abstract, methods: which University centre? I would either also give the city and country or delete this information in the abstract.

-) Abstract, methods: “with AMI” – please give the exact definition (see above).

-) Abstract, results: Please also state which troponin.

-) Methods: I suggest giving more information on how AMI and AF were diagnosed, even when you cite guidelines.

-) Methods: Were all patients treated at the ICU? This seems odd as you included all kinds of ACS, not only critically ill patients?

-) Methods: I find it problematic that you define NOAF as any AF episode lasting at least 30 seconds, this might be too short, please cite or describe how you came up with this time.

-) Methods: When you use the abbreviation PCI, I would use the correct full term for it (-intervention), not angioplasty.

-) Results, 3.4, predictors of NOAF: Please correct to “The rest OF THE parameters […]”.

-) Results, Table 5: Please state for which parameters the multivariate model was adjusted in the table legend.

-) Results, Table 6: The numbers of patients developing VT or VF seems quite high, what were the reasons for this? Also, I would not put VT and VF into any same group, as VF always also means CPR and VT does not (or do you mean pVT?).

-) Results, below Table 6, line 157: I would rephrase to “[…] surviving NOAF patients […]”.

-) Results, 3.6, lines 160-161: “was not able to obtain” – please revise this sentence grammatically.

-) Discussion, line 174: Please change to “from A high-volume tertiary care centre”.

-) Discussion, 4.1: Please change “the literature” to “literature” if you do not refer to specific literature.

-) Discussion, general: You list your findings and put them into context with recent literature. However, I miss the view on the real novelty your study provides (rather than a few new associations) – where is the “comprehensive” assessment of your list of markers of NOAF probability (you make this the main point of interest from the beginning on, even in the title)? This must be adapted to make the discussion (and also conclusion) interesting for a potential reader already knowing previous literature on the topic.

Author Response

Dear Reviewer,

As the authors of the article, we are very grateful for the reliable review and valuable comments on the paper's content, which allowed us to increase its scientific value. We would also like to thank you for the valuable time you devoted to reviewing our article. We have tried to follow all of the Reviewer's comments, trying to increase the value of our paper. We hope that we have met the Reviewer’s expectations regarding the revision of our article.

We want to refer to your specific comments:

Point 1: Title: Maybe shorten it to: “Comprehensive use of routine clinical parameters to identify patients at risk of new-onset atrial fibrillation in acute coronary syndrome” (or myocardial infarction, see comment #2)

Response 1: The authors wish to thank the Reviewer for raising that important point here. Due to the valuable Reviewers suggestion, we shorten the title to:  “Comprehensive use of routine clinical parameters to identify patients at risk of new-onset atrial fibrillation in acute myocardial infarction” (Page 1, lines 2 to 4).

Point 2: Please use the most up-to-date wording for acute coronary syndrome following the latest ESC guidelines.

Response 2: The authors wish to thank you for this valuable comment. In response to this comment, we revised the manuscript and tried to use the most up-to-date wording for the acute coronary syndrome. We should explain some discrepancies between up-to-date wording for ACS and using in our article “acute myocardial infarction (AMI)”. According to the ESC guidelines [1][2], ACS include patients with ST-segment elevation ACS and non-ST-segment elevation ACS (patients with typical non-ST-segment elevation myocardial infarction (NSTEMI) or unstable angina). In our study, we could include only patients with evidence of myocardial injury with necrosis in a clinical setting consistent with myocardial ischemia. Our study did not include patients with unstable angina – due to the retrospective character of the study our data was collected through MedStream Designer, where the patients with unstable angina were not precisely coded. Therefore, we decided to determine our patients of AMI, but not ACS. Due to the great clinical value of the Reviewer’s comment, we add this information to the limitations of the study page 11, lines 376 to 379) for further Reviewer’s acceptance.

Point 3: Abstract, 2nd sentence: This statement is very vague. I would either give more precise information or delete it.

Response 3: The authors wish to thank you for this valuable comment. We deleted this sentence.

Point 4: Abstract, methods: which University centre? I would either also give the city and country or delete this information in the abstract.

Response 4: We want to thank the Reviewer for this reliable comment. We added information about the University centre, city, and country (page 1, line 15; and page 2, line 68).

Point 5: Abstract, methods: “with AMI” – please give the exact definition (see above).

Response 5: The authors wish to thank you for this valuable comment. We added the exact definition to the abstract (page 1, lines 15 to 16). We hope that this section will be accepted by the Reviewer.

Point 6: Abstract, results: Please also state which troponin.

Response 6: The authors wish to thank for this valuable comment. We corrected and specified the troponin (page 1, line 19; page 4, line 150; page 6, line 170).

Point 7:  Methods: I suggest giving more information on how AMI and AF were diagnosed, even when you cite guidelines.

Response 7: Thank you very much for this important suggestion. We tried to add the appropriate information in the Methods section (page 2, lines 71 to 74, and 80 to 81).

Point 8:  Methods: Were all patients treated at the ICU? This seems odd as you included all kinds of ACS, not only critically ill patients?

Response 8: The authors wish to thank the Reviewer for this valuable question. In our clinical centre, we do not have coronary care unit only for ACS patients. However, every patient with AMI is admitted to the intensive cardiac care unit or monitored room in the regular cardiac ward, in order to be 24-h monitored. We add the appropriate explanation into the text (Methods section, page 2, lines 76 to 77). The authors hope that the Reviewer will accept this answer.

Point 9:  Methods: I find it problematic that you define NOAF as any AF episode lasting at least 30 seconds, this might be too short, please cite or describe how you came up with this time.

Response 9: The authors wish to thank for this valuable comment. We tried to diagnose atrial fibrillation according to the atrial fibrillation guidelines and be in concordance with the previous studies: the minimum duration of an ECG tracing of AF required to establish the diagnosis of clinical AF is at least 30 seconds or an entire 12-lead ECG. Any newly diagnosed AF, without any prior history of AF or atrial flutter, we considered as NOAF. We add the appropriate information into the Methods section (page 2, line 76 and line 82). The authors hope this answer will be acceptable for the Reviewer.

Point 10:  Methods: When you use the abbreviation PCI, I would use the correct full term for it (-intervention), not angioplasty.

Response 10: The authors wish to thank for this valuable comment. The abbreviations in the text and in tables are corrected (page 2, line 88; page 3, line 134).

Point 11: Results, 3.4, predictors of NOAF: Please correct to “The rest OF THE parameters […]”.

Response 11: Thank you very much for this remark. The suggested expression is used (page 5, line 165).

Point 12:  Results, Table 5: Please state for which parameters the multivariate model was adjusted in the table legend.

Response 12: The authors wish to thank the Reviewer for this valuable remark. Multivariate analysis was applied to all continuous data (dichotomized according to the cut-off values identified in ROC analyses) and categorical data which were significantly associated in the univariate analyses with NOAF (presented in the Figure 1). The multivariate model the set of variables accepted for the model was determined by the backward elimination method from the set of all these parameters (age, length of hospitalization, BNP, hsTnI, CRP, K, Hgb, leucocytes, N/L, TC, LDL, creatinine, LA size, and LVEF). We add this information in the Table 5 legend (page 7, lines 183 to 185. Some information was added in the Statistical method as well (page 3, lines 119 to 120). We hope this answer will be acceptable by the Reviewer.

Point 13: Results, Table 6: The numbers of patients developing VT or VF seems quite high, what were the reasons for this? Also, I would not put VT and VF into any same group, as VF always also means CPR and VT does not (or do you mean pVT?).

Response 13: We want to thank the Reviewer for this reliable comment. The authors fully agree with the Reviewer that the number of patients developing VT or VF seems quite high. This may be a consequence of including in this calculation not only patients with VT/VF within hospitalization but also at the moment of admission. Due to the retrospective character of our study, it was not possible to divide the time of malignant arrhythmia. In order to precisely feature the data, we add appropriate clarification in the limitations of the study (page 11, lines 388 to 390). The authors changed the Table 6, and divide VT/VF into two groups, for further Reviewers’ acceptance (page 7, lines 190 to 191).

Point 14:  Results, below Table 6, line 157: I would rephrase to “[…] surviving NOAF patients […]”.

Response 14: The authors wish to thank for this valuable comment. We changed the phrase to the correct one as suggested (page 7, line 192).

Point 15: Results, 3.6, lines 160-161: “was not able to obtain” – please revise this sentence grammatically.

Response 15: According to the Reviewers remark, the authors revised this sentence (Due to the retrospective character of the study, the information about pharmacological treatment for four patients was absent. Page 7, lines 195 to 196).

Point 16:  Discussion, line 174: Please change to “from A high-volume tertiary care centre”.

Response 16: Thank you very much for this remark, we followed it up. We changed the sentence to the correct one as suggested (page 8, lines 223).

Point 17: Discussion, 4.1: Please change “the literature” to “literature” if you do not refer to specific literature.

Response 17: The authors wish to thank for this valuable comment. We changed the phrase to the correct one as suggested (page 8, line 225).

Point 18:  Discussion, general: You list your findings and put them into context with recent literature. However, I miss the view on the real novelty your study provides (rather than a few new associations) – where is the “comprehensive” assessment of your list of markers of NOAF probability (you make this the main point of interest from the beginning on, even in the title)? This must be adapted to make the discussion (and also conclusion) interesting for a potential reader already knowing previous literature on the topic.

Response 18 The authors are very grateful to the Reviewer for this insightful remark. We tried to stress the novelty of our study in the abstract, introduction, and particularly in discussion (including the additional part “Novelties of the study”). 
Most of the previous studies in literature usually prioritized only one clinical parameter or prioritized one type of myocardial infarction (mostly STEMI), some studies were based on international registers, which, beyond the obvious advantages and vast numbers of participants, included data from different clinical centers, sometimes from other countries and even continents, which could implicate different diagnostic ways and different treatment possibilities. Moreover, most big registries in literature were based on the patients treated some years earlier, where the guidelines regarding acute coronary syndrome and our knowledge about atrial fibrillation were different. Therefore, we could say that our study is based on the nowadays population. Secondly, we could say that our AMI patients were treated in a modern way (with preferable invasive treatment): most patients (97%) had coronary angiography, and 82% of patients had percutaneous coronary intervention (PCI), in contrast to older studies. Next, we determined that in our nowadays patients with AMI, treated in a modern way, not only the single parameters, but many of routinely checked can help predict NOAF in AMI. We additionally calculated the cut-off values for those parameters, which could be helpful in everyday clinical practice. Furthermore, our study differs from the previous ones by a higher rate of triple antithrombotic treatment in NOAF patients.
The authors added appropriate changes in abstract (page 1, lines 13 and 24 to 25), introduction (page 1, lines 41 to 42), discussion (page 8, lines 218 to 222), conclusions (page 12, lines 396 and 399 to 400), and we add extra part 4.4. Novelties of the study (page 10 and 11, lines 344 to 372) trying to clarify our intention, for further Reviewer’s acceptance.

All changes to the manuscript text are tracked in the revised manuscript file with the Track Changes' mode.

We hope that you will find our paper interesting and you will approve it for publication.

Yours faithfully,

Prof. Ludmiła Daniłowicz-Szymanowicz, MD PhD
Monika Raczkowska-Golanko, MD

Round 2

Reviewer 2 Report

Dear authors, 

Thank you for addressing all my comments so extensively. I feel that the manuscript has improved substantially.